rsos.royalsocietypublishing.org

meteorology

land use, land surface temperature, impacts, remote sensing, geographic information system

**Author for correspondence:**
I. R. Orimoloye
e-mail: orimoloyeisrael@gmail.com

# Spatio-temporal analysis of land use dynamics and its potential indications on land surface temperature in Sokoto Metropolis, Nigeria

K. O. Ogunjobi[1], Y. Adamu[1], A. A. Akinsanola[1,2] and I. R. Orimoloye[3]

[1]Department of Meteorology and Climate Science, Federal University of Technology Akure, Nigeria
[2]School of Energy and Environment, City University of Hong Kong, Kowloon Tong, Hong Kong SAR, People's Republic of China
[3]Department of Geography and Environmental Science, University of Fort Hare, Private Bag X1314, Alice, 5700, Eastern Cape Province, South Africa

AAA, 0000-0002-0192-0082; IRO, 0000-0001-5058-2799

Land use change is the main driving force of global environmental change and is considered as most central to various debates on sustainable development. Even though a large volume of literature materials is available on land use/land cover change for many areas, very little work has been done on land use and its implications on land surface thermal characteristics over the Sokoto area of Nigeria, despite the strategic importance of the zone, including urbanization, increased population as well as the climate in the area, which is dominated by warm harmattan wind blowing Sahara dust inland. Thus, this study aimed at investigating the implications of urban growth on temporal variations of land surface temperature (LST) using remote sensing and geographic information system (GIS) techniques over Sokoto Metropolis, Nigeria between 1986 and 2016. The change detection of each land use class was carried out for each period using Landsat images obtained from the archives of the United States Geological Survey (USGS). The results revealed that the area has undergone a drastic transformation where built-up area witnessed changes at 10.77%, farmland and vegetation increased at the rate of 0.72% and 2.15%, respectively, for the period of study (1986–2016). While bare soil and water body decreased at the rate of 0.56% and 1.11%, respectively, during the study period. This shows that there exists a transformation from bare surface (desert) to vegetated surface especially between years 2009 and 2016.

rsos.royalsocietypublishing.org R. Soc. open sci. 5: 180661

The LST of Sokoto Metropolis was calculated from the satellite data, and the land surface temperature of each land use class was assessed for the study period. The maximum LST of Sokoto was 30.6°C, 32.8°C and 34.6°C for 1986, 1999 and 2016, respectively. This study has revealed the existence of a positive relationship between built-up area and LST over the area. This development might be as a result of anthropogenic activities through urban growth coupled with its potential impacts on urban climate. These are intensified by constant changes of the space, causing imbalance in the interactions between surface and atmosphere which may be extensively influenced or modified by various forms of land use.

# 1. Introduction

Urban climate investigations have for quite some times been concerned about the extent of the difference in ambient air temperature observation between urban communities and their surrounding rural areas, which collectively represent the urban heat events [1,2]. Owing to rapid rural–urban migration and population growth in the cities over the past decades, urbanization has taken place globally at an increased rate, and it is likely to continue in the subsequent decades [3–5]. Urbanization transforms the natural landscape into anthropogenic impervious surfaces which are land covers with buildings, roads, parking lots and other paved surfaces. As a result, one of the most significant environmental factors is the change of urban land surface temperature (LST) and atmospheric temperature, which in most cases significantly affects urban internal thermal characteristics [6,7]. Studies have also found that urban land development increased land surface thermal signal [7–9], which connotes that there exists a relationship between land use dynamics and land surface thermal characteristics. Liu & Weng [8] used landscape metrics to examine the relationship between land use and land change pattern and LST, and regarded landscape ecology as an effective tool to quantify the land use and LST patterns. The land use areas and landscape metrics can provide useful information for appraising and monitoring urban thermal environments in any given area [10–12].

Similarly, Nigeria has also witnessed an unprecedented growth in urbanization relative to the increasing population [13,14]. Studies have also shown that the population of the urban areas in Nigeria constitutes about 48.2% of the country's total population and projection has indicated that more than 60% will reside in urban centres by the year 2025 [15,16].

In Sokoto Metropolis of Nigeria, where this study is focused, land use and land cover patterns have undergone a rapid change due to accelerated expansion over the years. Urban growth has increased tremendously and extreme stress to the environment has occurred [17]. This increasing level of migration may be attributed to favourable socio-economic, agricultural, political and physical factors. Furthermore, environmental changes due to urbanization can have significant effects on local climate. One of the most familiar effects of urbanization is the urban heat island [17,18], which is the direct representation of environmental degradation. Increase in population and anthropogenic heat might have also contributed to this phenomenon [19,20]. Remote sensing and geographic information system (GIS) techniques have been identified as crucial ways of assessing urban heat phenomenon, and it had been widely used in monitoring urban growth and in detecting the changes that have occurred and its associated environmental injustice in the urban areas [7,21,22]. Hence, this study aimed at investigating implications of urban growth on temporal variations of land surface temperature using remote sensing and GIS techniques over Sokoto Metropolis, Nigeria between 1986 and 2016.

# 2. Study area, data and methodology

## 2.1. Study area

The study location is Sokoto Metropolis, Nigeria which is located in northwestern Nigeria (figure 1). The area is located at latitudes 13.04°–13.25°N and longitudes 5.1°–5.38°E. The area covers a land area of approximately 32 038 km$^2$ and its state shares borders with Niger Republic to the north, Zamfara and Katsina States to the east, Niger State to the southeast, Kwara State to the south and Benin Republic to the west. The southern boundary is arbitrarily defined by the Sudan savannah. Like the other parts of West Africa, the climate of the region is controlled largely by the two dominant air masses affecting the sub-region [23]. These are the dry, dusty, tropical-continental (cT) air mass (which

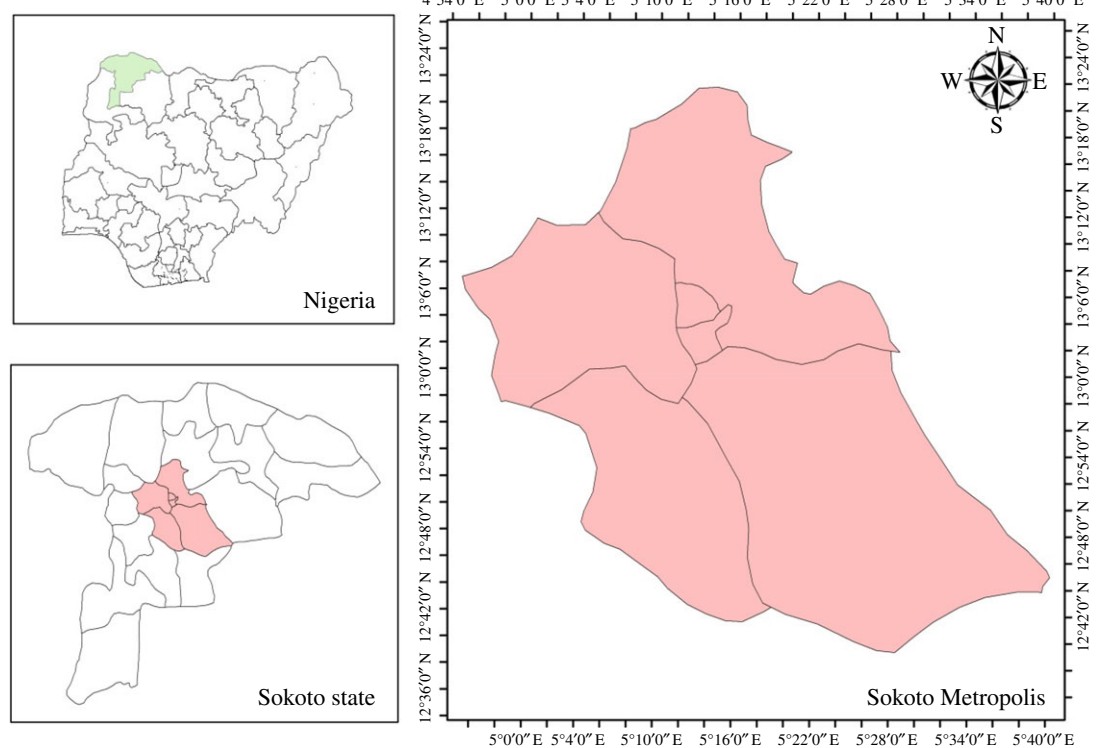

**Figure 1.** Map of the study area.

**Table 1.** Specification of satellite data used.

| date acquired | satellite | sensor | path/row | format |
| --- | --- | --- | --- | --- |
| 6 Jan 1986 | Landsat 5 | TM | 191/51 | GeoTIFF |
| 11 Nov 1999 | Landsat 7 | ETM+ | 191/51 | GeoTIFF |
| 24 Nov 2016 | Landsat 8 | OLI/TIRS | 191/51 | GeoTIFF |

originates from the Sahara region), and the warm, tropical-maritime (mT) air mass (which originates from the Atlantic Ocean). The annual rainfall for Sokoto ranges between 300 and 800 mm. The mean annual temperature is 34.5°C, although dry season temperatures in the region often exceed 40°C.

## 2.2. Data

The information in table 1 presents the satellite data used in this study which are: Landsat Thematic Mapper Imageries for year 1986, Enhanced Thematic Mapper for 1999 and OLI_TIRS image for 2016 obtained from the archives of United States Geological Survey (USGS).

## 2.3. Methodology

### 2.3.1. Image preprocessing

The Thematic Mapper (TM) for 1986, Enhanced Thematic Mapper (ETM+) for 1999 and operational land imager (OLI) and thermal infrared sensor (TIRS) for 2016 Landsat images acquired from USGS database were registered to 1 : 150 : 000 shapefile maps of Sokoto, Nigeria. Each image was radiometrically and geometrically corrected as used in Orimoloye et al. [7]. All the bands were used at a spatial resolution of 30 m, and these spectral bands are layer stacked to produce a composite image of the study area for each year (1986, 1999 and 2016) for the purpose of land use/land cover (LULC) classification image analysis. Thermal band 6 for Landsat 5 TM, ETM+ and band 10 for Landsat 8 TIRS were employed to calculate the LST from all the periods under consideration. The thermal bands have their original pixel sizes of 120 m for TM and 100 m for TIRS images which were resampled to 30 m using the

rsos.royalsocietypublishing.org    R. Soc. open sci. **5**: 180661

rsos.royalsocietypublishing.org R. Soc. open sci. **5**: 180661

nearest-neighbour algorithm to match the pixel size of other spectral bands. In order to examine the effects of human activities in the study area, a land cover classification is necessary for detection of LULC changes as a result of rapid urbanization from 1986 to 2016. The categories or classes considered are farmland, vegetation, water body, bare soil and built-up. After selecting training areas, a supervised classification with the maximum-likelihood algorithm was carried out to classify the Landsat images using bands 2 (green), 3 (red) and 4 (near infrared). Visual image interpretation was done with field knowledge and making reference to Google Earth images of the study area. The error matrixes of the three LULC maps were generated to assess the accuracy of the classification result.

### 2.3.2. Accuracy assessment

Land covers maps derived from the classification of images usually contain some sort of errors due to several factors that range from classification techniques to methods of satellite data capture. Therefore, evaluation of classification results is an important process in the classification procedure. The accuracy assessment was done by generating 200 equal random points for the classified images, using the accuracy assessment tool in Erdas Imagine. Study showed that the overall Kappa coefficient is 0.8798 (88%) while the overall accuracy or proportion classified is 89% which is above the usual benchmark of 85% [17].

### 2.3.3 Land surface temperature estimation

The following equation was used to convert the digital number (DN) of TIR bands of Landsat data into spectral radiance [7]:

$$L_\lambda = \left(\frac{\text{LMAX} - \text{LMIN}}{\text{QCALMAX} - \text{QCALMIN}}\right) * (\text{DN} - \text{QCALMIN}) + \text{LMIN} \tag{2.1}$$

where $L_\lambda$ is the spectral radiance at the sensor's aperture in W $(m^2\, sr\, \mu m)^{-1}$; QCAL is the quantized calibrated pixel value in DN; $\text{LMIN}_\lambda$ is the spectral radiance that is scaled to QCALMIN in W $(m^2\, sr\, \mu m)^{-1}$ $\text{LMAX}_\lambda$ is the spectral radiance that is scaled to QCALMAX in W $(m^2\, sr\, \mu m)^{-1}$; $\text{QCALMIN}_\lambda$ is the minimum quantized calibrated pixel value (corresponding to $\text{LMIN}_\lambda$); and $\text{QCALMAX}_\lambda$ is the maximum quantized calibrated pixel value (corresponding to $\text{LMAX}_\lambda$).

To convert the spectral radiance values to brightness temperature, equation (2.3) was used [7,20].

LST calculation, the digital numbers of the thermal bands of the satellite imagery were converted to spectral radiance using the bias and gain value. This conversion was obtained using the method used in [20],

$$C_{\text{VR}} = G(CV_{\text{DN}}) + B, \tag{2.2}$$

where $C_{\text{VR}}$ is the cell value radiance; $G$ is the gain value (0.05518 was used in this study); $CV_{\text{DN}}$ is the digital number (DN) of the band 6 imageries; and $B$ is the offset value (1.2378 was used in this study).

After the DNs for the thermal bands were converted to radiance values, the brightness temperatures were therefore, estimated as used in [20],

$$T_{\text{B}} = \frac{K_2}{\ln(((K_1 * \varepsilon / CVR) + 1))}, \tag{2.3}$$

where $T_{\text{B}}$ is an effective at-satellite temperature in K, $K_1 = 666.09$ is the constant used for this study, $K_2 = 1282.71$ is the constant used for this study, $\varepsilon = 0.92$ is the emissivity value used for correction which is appropriate for less vegetated areas, and $C_{\text{VR}}$ is the cell value inform of imageries obtained after converting the DNs to radiance.

Hence, the LST was estimated with the use of an emissivity-corrected land surface temperature's algorithm [7,20,24–26];

$$S_t(\text{K}) = \frac{T_{\text{B}}}{1 + (\lambda * T_{\text{B}}/\rho)\ln \varepsilon}, \tag{2.4}$$

where; $S_t$ is the emissivity corrected land surface temperature in K; $T_{\text{B}}$ is the satellite brightness temperature in K, retrieved previously; $\lambda = 11.5\, \mu m$;

$$\rho = \frac{h * c}{\delta} = 1.438 \times 10^{-2}\, m\, K = 1.438 \times 10^{-8}\, \mu m\, K;$$

rsos.royalsocietypublishing.org    R. Soc. open sci. **5**: 180661

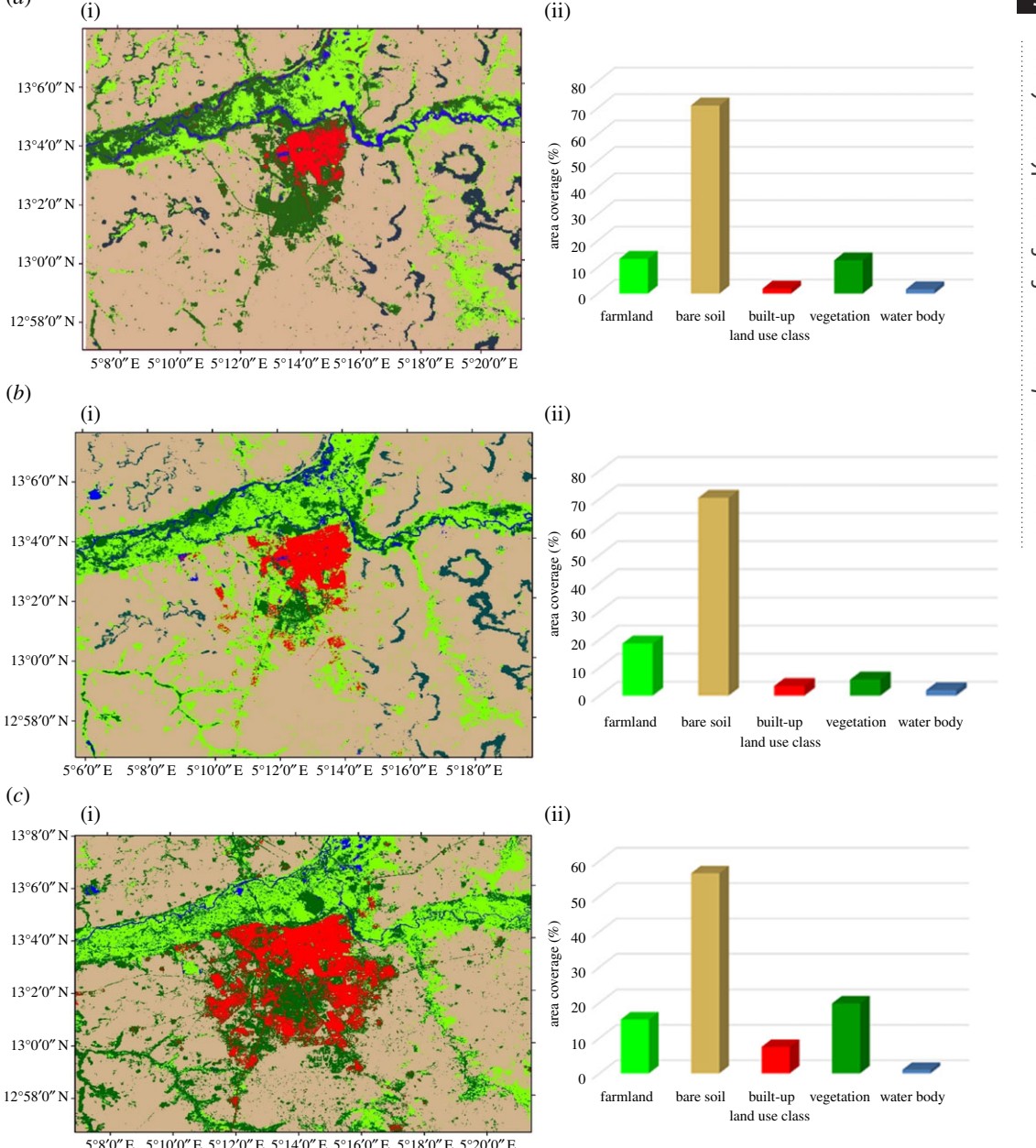

**Figure 2.** Land use land cover classification for (*a*) 1986, (*b*) 1999, (*c*) 2016.

$h$ is the Planck's constant = $6.626 \times 10^{-34}$ J s$^{-1}$; $c$ is the velocity of light = $2.998 \times 10^{8}$ m s$^{-1}$; and $\delta$ is the Boltzman's constant = $1.38 \times 10^{-23}$ J k$^{-1}$.

The final land surface temperature in °C was calculated with the following equation:

$$S_t(°C) = S_t(K) - 273.15.$$

(2.5)

# 3. Results and discussion

## 3.1. Assessment of land use land cover dynamics

The land use land cover of Sokoto was mapped from Landsat imageries for 1986, 1999 and 2016 as presented in figure 2. The land use was classified into five categories which include farmland, bare soil, built-up, vegetation and water body. In the year 1986, the result shows that the bare soil occupies the majority of the land use area with about 71% (figure 2*a*), followed by farmland with 13% while vegetation, built-up and water body cover areas of about 12.3%, 1.8% and 1.6%, respectively.

**Table 2.** Change detection of land use during the study periods.

| land cover class | change in area (hectares in thousands) | percentage change (%) | mean area change (hectares in thousands/year) | mean percentage change (%/year) |
|---|---|---|---|---|
| 1986 – 1999 | | | | |
| farmland | 2.96 | 42.60 | 0.23 | 3.28 |
| bare soil | −0.09 | −0.24 | −0.01 | −0.02 |
| built-up | 0.85 | 87.07 | 0.07 | 6.70 |
| vegetation | −3.62 | −54.29 | −0.28 | −4.18 |
| water body | 0.18 | 22.15 | 0.01 | 1.70 |
| 1999 – 2016 | | | | |
| farmland | −1.46 | −14.74 | −0.09 | −0.87 |
| bare soil | −6.22 | −16.55 | −0.37 | −0.97 |
| built-up | 2.32 | 126.21 | 0.14 | 7.42 |
| vegetation | 7.93 | 260.07 | 0.47 | 15.30 |
| water body | −0.46 | −45.46 | −0.03 | −2.67 |
| 1986 – 2016 | | | | |
| farmland | 1.50 | 21.58 | 0.05 | 0.72 |
| bare soil | −6.32 | −16.76 | −0.21 | −0.56 |
| built-up | 3.17 | 323.18 | 0.11 | 10.77 |
| vegetation | 4.31 | 64.59 | 0.14 | 2.15 |
| water body | −0.28 | −33.38 | −0.01 | −1.11 |

In the year 1999, the information in figure 2b reveals that bare soil occupied majority of the land use area with area coverage of about 70%, farmland increased to about 18.5% of the area coverage, while vegetation decreased from about 12.5% to 5.7%. The result further reveals that built-up and water body witnessed increment with about 3.4% and 1.9%, respectively, during the same period, which connotes that the vegetation and other land use features have been influenced by built-up and development in the area. In the most recent year (2016), even though there was a decrease in the bare soil land use area coverage from 70% to 56.5%, the land use land cover around Sokoto was still majorly bare soil (figure 2c). Farmland, built-up, vegetation and water body occupied 13.2%, 7.4%, 19.7% and 1% of the total area coverage of the land use land cover for the same period.

The information in table 2 reveals the rate of change during the period of study for each time interval. Firstly, the changes between year 1986 and 1999: built-up has the highest change with a mean percentage change of 6.7% change; farmland has the rate of 3.28% change during the same period. Vegetation and bare soil changed at the rate of −4.18% and 0.02%, respectively, during the same period. Secondly, from 1999 to 2016, farmland, bare soil and water body decreased at the rate 0.87%, 0.97% and 2.67%, respectively, while built-up and vegetation increased at the rate of 7.42% and 15.3% change during the period. Lastly, change that occurred for the 30-year period of investigation, 1986 to 2016: built-up changed the most at 10.77% change for the years of investigation. Farmland and vegetation increased at the rate of 0.72% and 2.15% change, respectively, while bare soil and water body decreased at the rate of 0.56% and 1.11% change, respectively, during the study period.

## 3.2. Spatial distribution of land surface temperature between 1986 and 2016

The information in figure 3 shows the land surface temperature (LST) estimated from Landsat images for 1986, 1999 and 2016. It represents the spatial distribution of LST across various land use features. In 1986 (figure 3a), the maximum land surface temperature was 34.1°C, minimum was 20.4°C with a mean value of 30.6°C while in 1999 (figure 3b), the maximum land surface temperature was 36.9°C, minimum was 21.3°C with a mean value of 32.8°C. In 2016 (figure 3c), the maximum land surface temperature was 38.6°C, minimum was 24.1°C with a mean value of 34.6°C. The land surface temperature distribution shows a spatial variability across the study area. From the spatial distribution of land surface

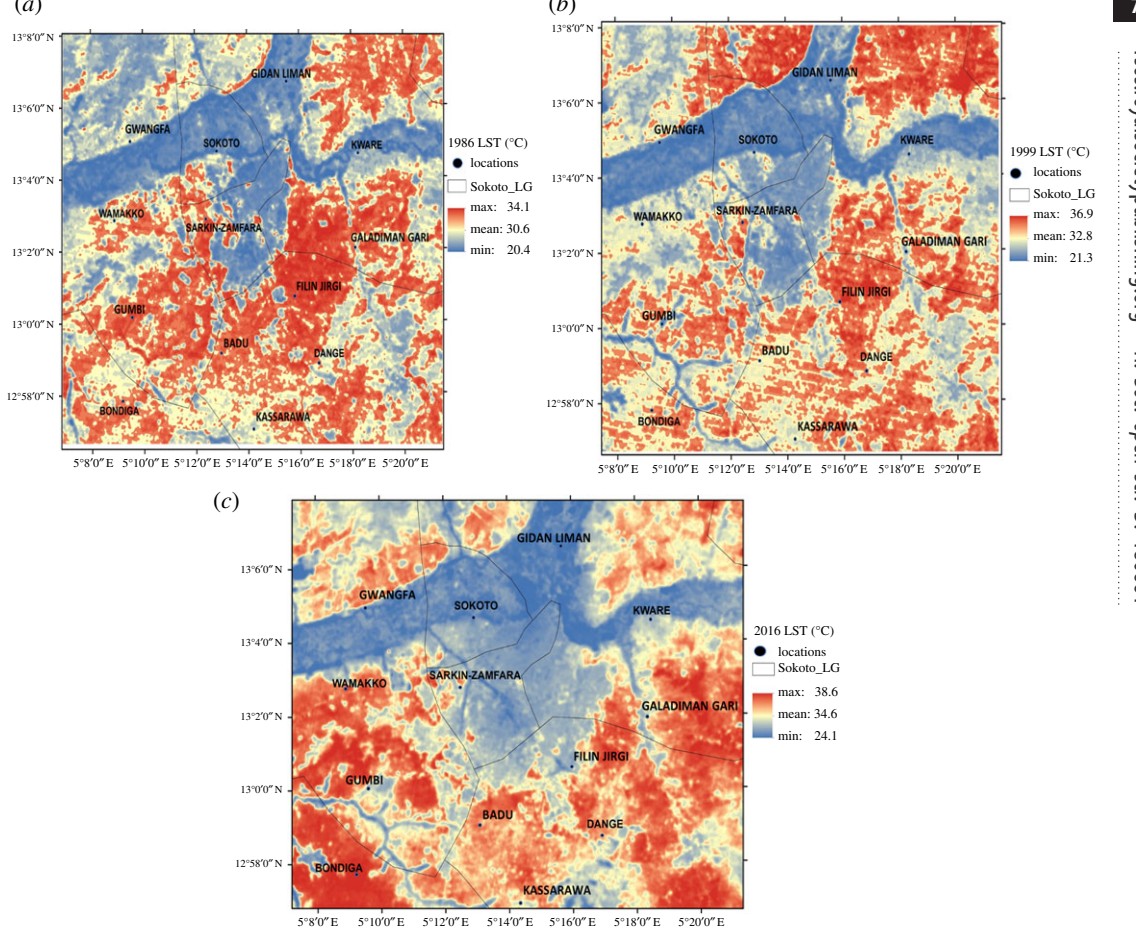

**Figure 3.** Land surface temperature (°C) over Sokoto for (a) 1986, (b) 1999, (c) 2016.

temperature for the periods of study, the maximum LST was around the bare soil area, while the minimum LST was around the vegetated and water surfaces. The LST around the built-up area was lower than that of the surrounding bare soil which could be as a result of the climatic zone of the study area. Sokoto lies in the Sahel region and the bare soil around Sokoto is an offshoot of the desert area around the Sahel. It was also observed that as the area covered with built-up area increased, there was a corresponding increase in the area coverage of LST outwards from the urban area, this development might be as a result of heat generated in the built area as asserted by previous studies [7,20,28].

The information in figure 4 shows the distribution of the mean land surface temperature over the various land use classes for Sokoto in 1986. The highest mean land surface temperature of 30.8°C was over the bare soil followed by built-up area with mean land surface temperature of 29.4°C which connotes that bare soil incorporated with urban development might have significantly influenced land surface thermal characteristics in the area [7,27]. The mean LSTs a for farmland and vegetation were 27.9°C and 26.7°C, respectively, while water body had the lowest mean LST with a value of 26.3°C. A similar result was observed in 1999 and 2016 (figure 4). The mean LSTs for bare soil and built-up were 30.1°C and 31.9°C, respectively, in 1999, while those of farmland, vegetation and water body were 30.4°C, 28.2°C and 27.7°C, respectively. More so, the mean LSTs for bare soil and built-up were 36.7°C and 32.5°C, respectively, in 2016, while those of farmland, vegetation and water body were 31.2°C, 29.03°C and 29°C, respectively (figure 4).

# 4. Summary and conclusion

This study has presented a spatio-temporal analysis of land use land changes and its potential impacts on land surface thermal characteristics over the 30-year period between 1986 and 2016. The results have shown that bare soil occupied the majority of the total area coverage of Sokoto Metropolis with values

rsos.royalsocietypublishing.org    R. Soc. open sci. 5: 180661

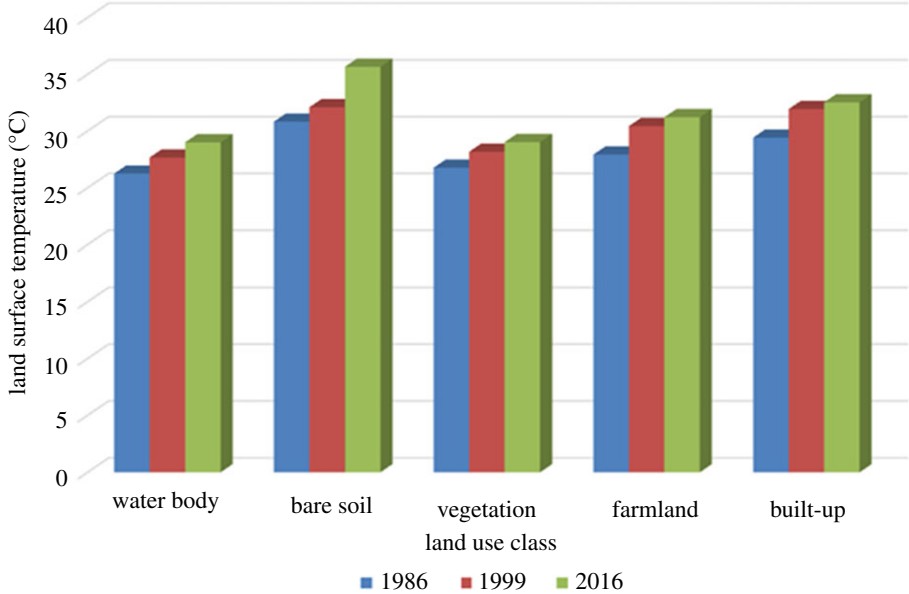

**Figure 4.** Distribution of land surface temperature over land use/cover classes for 1986, 1999 and 2016.

of 71%, 70% and 56.5% for 1986, 1999 and 2016, respectively. The urban (built-up) area, on the other hand, occupied a small section of the total area coverage of 1.8% in 1986 built with the increase in population and human activities outwards from the urban centre, the area coverage of built-up area increased to 3.4% of the total land use area cover. The built-up area further increased in 2016 to 7.48% of the total coverage area. The increase in the urban (built-up) area is expected to be due to the increase in the general population of Sokoto Metropolis. With the increase in population and human activities, the resulting effect is increased pressure on land surface, which results in the modification of the natural surfaces from vegetative covers to impervious surfaces, vegetative covers to farmlands, farmlands to impervious surfaces, which have impacted extensively on land surface thermal signals [7,27,29]. If this development persists, it may have adverse effects on the environment and the health of the residents of the area [7,30,31].

The result has further revealed that there exists a drastic change in bare soil and built-up area: while bare soil decreased throughout the study periods, built-up increased all through, and this has contributed immensely to the development of urban climate in the area [32–35]. The other land cover classes revealed variation in the change for the years such as vegetation which decreased from 6.67 to 3.05 hectares in thousands from 1986 to 1999, which connotes that that study area has undergone notable development, which in turn has also altered LST during the period. The statistics of the change detection was also carried out, which revealed that built-up area varied more compared to the other land features from year to year with a mean percentage change of 10.77% during the period (1986–2016). The land surface temperature of Sokoto Metropolis was estimated from the Landsat imageries for 1986, 1999 and 2016. The mean land surface temperature was 30.6°C, 32.8°C and 34.6°C for 1986, 1999 and 2016, respectively. It was observed that LST varied spatially for all the images. This may be as a result of different land covers of the natural landscape. The land surface partitions the LST differently based on the land cover and their levels of impact on LST. The mean values of LST were extracted from the land cover classes and plotted in order to reveal how land surface temperature varied and was influenced by land use features (figure 4). It was observed that bare soil had the highest values of LST followed by built-up area while water body had the lowest LST value throughout the period of study, as identified in prior studies which signify that urban growth has altered natural land surface thermal characteristics in the area [7,36,37]. In order to assess the LST change over the years for each land cover class, the mean LST of each class was plotted and revealed an increase in the mean LST of each land feature (figure 4). Hence, the result from the study has revealed that land use dynamics had modified land surface temperature of the area during the period.

The study also further revealed that the increase in LST in built-up and bare surface areas might be as a result of the presence of dark surfaces such as roadways and rooftops efficiently absorbing heat from sunlight and reradiating it as thermal infrared radiation; these surfaces can reach temperatures of 50–70°F (28–39°C) higher than surrounding air [35,37]. Also, urban areas are relatively devoid of

rsos.royalsocietypublishing.org R. Soc. open sci. 5: 180661

vegetation, especially trees that would have provided shade and cool the air through evapotranspiration, which has been altered by built-up environment [33,38,39]. As cities grow, the heat island effect expands both in extent and intensity [38,39]. This is especially true if the pattern of development features extensive tree-cutting as well as road construction as a result of human activities of deforestation and residential purposes [7,27,40–42].

The present study has revealed the current state of land use dynamic and its potential implications on land surface thermal characteristics over the study area as well as the influence of human activities on the natural landscape, particularly the built-up areas which increased over time due to population increase and recent development in the region. This basically shows an increase in the land surface temperature of Sokoto Metropolis. This study has shown that LST for water body from 1986 to 2016 increased from 26.3°C to 29°C, bare soil increased from 30.8°C to 35.7°C, vegetation increased from 26.8°C to 29.03°C, farmland increased from 27.9°C to 31.2°C and built-up increased from 29.4°C to 32.5°C during the same period. The study further revealed that built-up area witnessed increase compared to other land features and the change was a continual increase in the area coverage. The LST was also assessed with an increase in the mean LST of each land cover class. The implication of the increase in area coverage of built-up area is that of modification of the local climate around the Sokoto Metropolis. The increase in LST also has its implications in farming due to increase in the temperature over the soil cover. It is, therefore, necessary to monitor the land surface thermal signals variation which can be very useful information for various sectors such as agriculture, health and the environment.

Data accessibility. Datasets used in this study obtained from the archives of United States Geological Survey (USGS). https://earthexplorer.usgs.gov/.

Authors' contributions. All authors contributed immensely to the manuscript. K.O.O. supervised the research, Y.A. analysed the data. A.A.A. and I.R.O. organized, validated and wrote the paper.

Competing interests. The authors declare that there is no conflict of interests regarding the publication of this paper.

Funding. Not applicable.

Acknowledgements. The authors are very grateful to United States Geological Survey (USGS) for proving Landsat images.

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
