## [Reviewer comments · Royal Society Open Science]

Review History

RSOS-180661.R0 (Original submission)

Review form: Reviewer 1 (Alexander Solomon)

Is the manuscript scientifically sound in its present form?

Yes

Are the interpretations and conclusions justified by the results?

Yes

Is the language acceptable?

Yes

Is it clear how to access all supporting data?

Yes

Do you have any ethical concerns with this paper?

No

Have you any concerns about statistical analyses in this paper?

No

Recommendation?

Accept with minor revision (please list in comments)

Comments to the Author(s)

The manuscript is well written, has an important environmental message, and should be of great interest to the readers. However, there are a couple of other minor issues that I mention in the "comments to the authors below" (Appendix A). Overall, it is an important study, and should be considered for publication in RSOS, once the minor correction and adjustment have been resolved.

Best regards

Review form: Reviewer 2

Is the manuscript scientifically sound in its present form?

Yes

Are the interpretations and conclusions justified by the results?

Yes

Is the language acceptable?

Yes

Is it clear how to access all supporting data?

Yes

Do you have any ethical concerns with this paper?

No

Have you any concerns about statistical analyses in this paper?

No

Recommendation?

Major revision is needed (please make suggestions in comments)

Comments to the Author(s)

Please summarize the highlights of your study

Decision letter (RSOS-180661.R0)

29-Aug-2018

Dear Dr Orimoloye,

The editors assigned to your paper ("Spatio-temporal Analysis of Land Use Dynamics and its Potential Indications on Land Surface Temperature in Sokoto Metropolis, Nigeria.") have now received comments from reviewers. We would like you to revise your paper in accordance with the referee and Associate Editor suggestions which can be found below (not including confidential reports to the Editor). Please note this decision does not guarantee eventual acceptance.

Please submit a copy of your revised paper before 21-Sep-2018. Please note that the revision deadline will expire at 00.00am on this date. If we do not hear from you within this time then it will be assumed that the paper has been withdrawn. In exceptional circumstances, extensions may be possible if agreed with the Editorial Office in advance. We do not allow multiple rounds of revision so we urge you to make every effort to fully address all of the comments at this stage. If deemed necessary by the Editors, your manuscript will be sent back to one or more of the original reviewers for assessment. If the original reviewers are not available, we may invite new reviewers.

- Data accessibility

If you wish to submit your supporting data or code to Dryad (<http://datadryad.org/>), or modify your current submission to dryad, please use the following link:
<http://datadryad.org/submit?journalID=RSOS&manu=RSOS-180661>

- **Competing interests**

- **Authors' contributions**

- **Acknowledgements**

- **Funding statement**

Please note that Royal Society Open Science charge article processing charges for all new submissions that are accepted for publication. Charges will also apply to papers transferred to Royal Society Open Science from other Royal Society Publishing journals, as well as papers submitted as part of our collaboration with the Royal Society of Chemistry (<http://rsos.royalsocietypublishing.org/chemistry>). If your manuscript is newly submitted and subsequently accepted for publication, you will be asked to pay the article processing charge, unless you request a waiver and this is approved by Royal Society Publishing. You can find out more about the charges at <http://rsos.royalsocietypublishing.org/page/charges>. Should you have any queries, please contact openscience@royalsociety.org.

Kind regards,

Royal Society Open Science Editorial Office
Royal Society Open Science
openscience@royalsociety.org

on behalf of Dr Pablo Gonzalez (Associate Editor) and Prof. Jon Blundy (Subject Editor)

Associate Editor's comments (Dr Pablo Gonzalez):

Dear Authors,

First of all, apologizes for the long time it took to review your manuscript. Now, we have two reviewing reports and I am confident they will be of great help to improve the manuscript. At this point, I consider you have to make significant changes but overall, they can be addressed within the limits of a major revision.

Looking forward to your revised manuscript. Please address all comments before resubmitting the manuscript to RSOS.

Kind regards,
Pablo J Gonzalez

Comments to Author:

Reviewers' Comments to Author:

Reviewer: 1

Comments to the Author(s)

The manuscript is well written, has an important environmental message, and should be of great interest to the readers. However, there are a couple of other minor issues that I mention in the "comments to the authors below". Overall, it is an important study, and should be considered for publication in RSOS, once the minor correction and adjustment have been resolved.

Best regards

Reviewer: 2

Comments to the Author(s)

Please summarize the highlights of your study

Author's Response to Decision Letter for (RSOS-180661.R0)

See Appendix B.

Decision letter (RSOS-180661.R1)

10-Oct-2018

Dear Dr Orimoloye:

Manuscript ID RSOS-180661.R1 entitled "Spatio-temporal Analysis of Land Use Dynamics and its Potential Indications on Land Surface Temperature in Sokoto Metropolis, Nigeria." which you

submitted to Royal Society Open Science, has been reviewed. The comments of the reviewer(s) are included at the bottom of this letter.

Please submit a copy of your revised paper before 02-Nov-2018. Please note that the revision deadline will expire at 00.00am on this date. If we do not hear from you within this time then it will be assumed that the paper has been withdrawn. In exceptional circumstances, extensions may be possible if agreed with the Editorial Office in advance. We do not allow multiple rounds of revision so we urge you to make every effort to fully address all of the comments at this stage. If deemed necessary by the Editors, your manuscript will be sent back to one or more of the original reviewers for assessment. If the original reviewers are not available we may invite new reviewers.

- Ethics statement

- Data accessibility

- Competing interests

- Authors' contributions

All submissions, other than those with a single author, must include an Authors' Contributions section which individually lists the specific contribution of each author. The list of Authors should meet all of the following criteria; 1) substantial contributions to conception and design, or

acquisition of data, or analysis and interpretation of data; 2) drafting the article or revising it critically for important intellectual content; and 3) final approval of the version to be published.

- Acknowledgements

- Funding statement

Please note that Royal Society Open Science charge article processing charges for all new submissions that are accepted for publication. Charges will also apply to papers transferred to Royal Society Open Science from other Royal Society Publishing journals, as well as papers submitted as part of our collaboration with the Royal Society of Chemistry (<http://rsos.royalsocietypublishing.org/chemistry>). If your manuscript is newly submitted and subsequently accepted for publication, you will be asked to pay the article processing charge, unless you request a waiver and this is approved by Royal Society Publishing. You can find out more about the charges at <http://rsos.royalsocietypublishing.org/page/charges>. Should you have any queries, please contact openscience@royalsociety.org.

on behalf of Dr Pablo Gonzalez (Associate Editor) and Prof. Jon Blundy (Subject Editor)
openscience@royalsociety.org

Associate Editor Comments to Author (Dr Pablo Gonzalez):

Dear authors,

Thanks for your effort on revisiting the text. I see that the manuscript has improved by taking into account the reviewers comments.

However, from my own read of the manuscript, I found that some points need clarification. I would like to give you the opportunity to streamline them and clarify the following points, before accepting it. I have focused on the abstract, but if you want to address them it might need to add some changes in the maintext as well.

Abstract

- Line 19: Please give some details of what type of strategic importance in a wider context for the Sokoto region. e.g., Is it its vegetation type, urban growth model,... etc?
- Line 20: Spell out the LST (Land Surface Temperature). It is the first time that appear in the text
- Line 21: Same for GIS
- Line 26: Are the decrease of bare surface and water body in per year units? Please clarify.
- In addition, Are changes less than 2% statistically significant? My first impression is that they small compared with the method of analysis/classification and the size of the area analyzed. Please provide an error estimation, and/or revisit the abstract text for those land cover types.
- Finally, what could be a reason for the positive correlation between urban growth and LST?

Figure 3b seems to be affected by some Landsat sensor stripping effect. Is that the case? That might compromise the LST estimation to be compared with 1986 and 2016 images.

Author's Response to Decision Letter for (RSOS-180661.R1)

See Appendix C.

Decision letter (RSOS-180661.R2)

09-Nov-2018

Dear Dr Orimoloye,

I am pleased to inform you that your manuscript entitled "Spatio-temporal Analysis of Land Use Dynamics and its Potential Indications on Land Surface Temperature in Sokoto Metropolis, Nigeria." is now accepted for publication in Royal Society Open Science.

Kind regards,
Andrew Dunn

Royal Society Open Science Editorial Office
Royal Society Open Science
openscience@royalsociety.org

on behalf of Dr Pablo Gonzalez (Associate Editor) and Prof. Jon Blundy (Subject Editor)
openscience@royalsociety.org

Appendix A

Reviewer' comments on Article "RSOS-180661": comments to the editor and authors

Topic: Spatio-temporal Analysis of Land Use Dynamics and its Potential Indications on Land Surface Temperature in Sokoto Metropolis, Nigeria

INTEREST TO READERS: High

ORIGINALITY AND CONTENT OF NEW INFORMATION: High

STUDY DESIGN: Adequate

ANALYSES: Appropriate

VALIDITY OF CONCLUSIONS: Valid

CLARITY OF WRITING: High

RECOMMENDATIONS: ACCEPT

IF RECOMMEND ACCEPTANCE: is the **length** appropriate? : Yes

Are there any **figures or tables** that are unnecessary: No

COMMENTS TO THE EDITORS AND AUTHORS:

The authors present interesting information on natural landscapes dynamics and its potential implication of land surface temperature (1986 – 2016) with emphasis on land use land change. Their study assessed the land surface temperature and it found varied spatially for all the images as a result of different land covers of the land surface with their different thermal characteristics. This study has revealed an existence of negative relationship between built-up area and land surface temperature over the area.

Result further revealed that there exists a drastic change in change in bare soil and built-up area, while bare soil decreased throughout the study periods and built-up increased all through and have contributed immensely to the development of urban climate in the area.

The manuscript is well written, has an important environmental message, and should be of great interest to the readers. However, there are a couple of other minor issues that I mention in the "comments to the authors below". Overall, it is an important study, and should be considered for publication in RSOS, once the minor correction and adjustment have been resolved.

Abstract

Well written, but author should rephrase long sentences

Introduction

I will suggest that author should revise the introduction section and pay more attention to use of punctuation. More so, author should rephrase long sentences in the text.

Methods

In accuracy assessment section more still need to be done as the accuracy and omission errors were not well presented. Author should revise and gives the percentage of the accuracies.

Result and Discussion

Pg. 13 line 33, remove at after rate and replace it accordingly.

Conclusion

Appropriate

Good luck

Appendix B

University of Fort Hare
Together in Excellence

Geography and Environmental Science
University of Fort Hare
Private Bag X1314, Alice 5700
Eastern Cape
Republic of South Africa
Email: iorimoloye@ufh.ac.za
orimoloyeisrael@gmail.com
Mobile: +27732244901, +2348060221246
Skype : orimoloyeisrael@yahoo.com

Dear Editor,

We are pleased to submit the revised version of our manuscript titled “**Spatio-temporal Analysis of Land Use Dynamics and its Potential Indications on Land Surface Temperature in Sokoto Metropolis, Nigeria**” (Ref: RSOS-180661). We appreciate the time and efforts of the editor and anonymous reviewers in reviewing the manuscript. We have addressed carefully all the comments in the review report as highlighted in the table below and also in the text (highlighted yellow) and believed that the revised version can meet the journal publication requirement.

Details on the revised version are outlined in the table below:

Reviewer #1: The authors are grateful to the Reviewer for the time spent in analyzing the manuscript, and for the constructive criticisms that have helped us to improve its quality. We are happy the reviewer finds our work interesting and, eventually worthy to be published.

Points	Reviewer Comments	Author's Response	Page
Abstract			
Point 1	Well written, but author should rephrase long sentences	Addressed	Pg. 1 lines 16-20, 23-27
Introduction			
Point 2	I will suggest that author should revise the introduction section and pay more attention to use of punctuation. More so, author should rephrase long sentences in the text.	Addressed:	Pg. 2 lines 38, 44, and 46. Pg. 3 line 61
Methods			
Pont 3	In accuracy assessment section more still need to be done as the accuracy and omission errors were not well presented. Author should revise and gives the percentage of the accuracies	Addressed as suggested	Pg. 6, lines 117 to 119
Results and discussion			
Point 4	Pg. 13 line 33, remove at after rate and replace it accordingly.	Addressed	Pg. 12 line 183

Reviewer 2#: The authors are grateful to the reviewer for the time spent in analyzing the manuscript, and for all the constructive comments that have helped us to improve its quality. We are happy the reviewer finds our work interesting and, eventually worthy to be published

Points	Reviewer Comments	Author's Response	
Point 1		Addressed. After the estimation of brightness temperature, we calculated the LST for study area using method used in Orimoloye et al., 2018. We have addressed it in the text as suggested by the reviewer	Pg. 8 lines 142-143
Point 2	The author should supplement the pre-processing steps before calculating the land surface temperature, which is very important for the inversion of land surface temperature	Addressed	Pg. 6 lines 102-105, pg. 8 lines 151, 159
Point 3	If land surface temperature calculation method in the manuscript is not the author's innovation, please add references	Addressed by the authors as suggested by the reviewer, some of the references are 7, 24, 25, 26 and 30	Pg. 7 lines 146
Point 4	Land surface temperature retrieved by remote sensing data is generally different from the real land surface temperature, it is better for the author to verify the data with the measured data or the references available	The issues raised by the reviewer is very important, but in this case, we used only remotely sensed information to assess LST of the study area not integration of meteorological observation and remote sensing data. However, our future research we consider the issues raised by the reviewer as time and data accessibility may fail us to include that in the present research. More so, we noticed that most studies on LST estimation used remotely sensed data, and	

		we cited some of them in the present study.	
Pont 5	I think both Figure 4 and Figure 5 in the manuscript show LST of different land use types. So, there is no difference between the two. Please explain the necessity of the coexistence of Figure 4 and Figure 5 in the manuscript, otherwise please modify it.	Thank you for the observation. We removed figure 5 from the text to avoid repetition	Pg. 18 line 232
Point 6	There are some grammatical problems and formatting problems in the article, for example, in the second paragraph of the introduction, “ In the near future, it is expected that the global rate of urbanization will increase the world urban ”; meanwhile, the last sentence of the abstract is not consistent with the font. Please check the manuscript carefully	We addressed the issues raised by the reviewer and the manuscript has been proofread by professional language editor. More so, we formatted the manuscript to one font (Times New Roman 12).	Pgs. 2-3 lines 50-53
Pont 7	In the introduction section, the author uses many sentences to introduce land surface temperature and land use. The research status of the relationship between land surface temperature and land use is relatively lacking. Please add relevant literature	Addressed as suggested by the reviewer.	Pg. 2 lines 42-51

We believe the revised manuscript has met all the requirements from both reviewers.

Sincere regards

Dr. Orimoloye I. R

Appendix C

Dear Editor,

We are pleased to submit the revised version of our manuscript titled “**Spatio-temporal Analysis of Land Use Dynamics and its Potential Indications on Land Surface Temperature in Sokoto Metropolis, Nigeria**” (Ref: RSOS-180661). We appreciate the time and efforts of the editor and reviewers in reviewing the revised version of the manuscript. We have addressed carefully all the comments in the review report as highlighted in the table below and also in the text (highlighted yellow) and believed that the latest revised version can meet the journal publication requirement.

Details on the revised version are outlined in the table below:

Reviewer #1: The authors are grateful to the Reviewer for the time spent in analyzing the manuscript, and for the constructive criticisms that have helped us to improve its quality. We are happy the reviewer finds our work interesting and, eventually worthy to be published.

Points	Reviewer Comments	Author's Response	Page
Abstract			
Point 1	Line 19: Please give some details of what type of strategic importance in a wider context for the Sokoto region. e.g., Is it its vegetation type, urban growth model,... etc ?	Addressed	Pg. 1 lines 19-21, 23-27
2	Line 20: Spell out the LST (Land Surface Temperature). It is the first time that appear in the text	Addressed as suggested	Pg. 1 line 22
3	Line 21: Same for GIS	Addressed as suggested	Pg. 1 line 23
4	Line 26: Are the decrease of bare surface and water body in per year units? Please clarify.	Addressed. Thank you for calling our attention to that, that change is for the period of study not per year	Pg. 1 line 28, pg. 13 lines 198 and 206
5	In addition, Are changes less than 2% statistically significant? My first impression is that they small compared with the method of analysis/classification and the size of the area analyzed. Please provide an error estimation, and/or revisit the abstract text for those land cover types.	Thank you for this wonderful suggestion. We could not perform any statistical significant test because of the amount of sample sizes. But we have included briefly the level of accuracy and bias/error in the text. More so, whether 2 % is statistically significant or not, we presented the	

		outcome of the analysis, but 1 % is statistically significant as reported in previous related studies. For further study in line with the present study, we will consider in the next research by increasing the size of our sample size.	
6	Finally, what could be a reason for the positive correlation between urban growth and LST?	Addressed	Pg. 1 lines 35-38
7	Figure 3b seems to be affected by some Landsat sensor stripping effect. Is that the case? That might compromise the LST estimation to be compared with 1986 and 2016 images	All the satellite images used for this study are less than 10% cloud cover (recommended) and they are free of atmospheric disturbance	

We believe the revised manuscript has met all the requirements from the reviewers.

Sincere regards

Dr. Orimoloye I. R